

# Effective field theory of a vortex lattice in a bosonic superfluid

**Sergej Moroz[1], Carlos Hoyos[2], Claudio Benzoni[1] and Dam Thanh Son[3]**

**1** Physik-Department, Technische Universität München,
D-85748 Garching, Germany
**2** Department of Physics, Universidad de Oviedo,
c/ Federico Garcia Lorca 18, 33007, Oviedo, Spain
**3** Kadanoff Center for Theoretical Physics, University of Chicago,
Chicago, IL 60637 USA

## Abstract

Using boson-vortex duality, we formulate a low-energy effective theory of a two-dimensional vortex lattice in a bosonic Galilean-invariant compressible superfluid. The excitation spectrum contains a gapped Kohn mode and an elliptically polarized Tkachenko mode that has quadratic dispersion relation at low momenta. External rotation breaks parity and time-reversal symmetries and gives rise to Hall responses. We extract the particle number current and stress tensor linear responses and investigate the relations between them that follow from Galilean symmetry. We argue that elementary particles and vortices do not couple to the spin connection which suggests that the Hall viscosity at zero frequency and momentum vanishes in a vortex lattice.



# 1   Introduction

Since the discovery of superfluidity in $^4$He, superfluids provide a never-ending source of inspiration for experimental and theoretical research in low-energy physics. Although a regular superfluid flow is necessarily irrotational, superfluids can carry finite angular momentum in the form of topological defects known as quantum vortices, which nucleate naturally in response to external rotation. Under slow rotation, the density of bosons is much larger than that of the topological defects and the quantum vortices form a regular vortex lattice, which has been observed in superfluid He [1] and more recently also in cold atomic BECs [2]. At larger rotation frequencies, the vortex cores start to overlap, and at a certain point the vortex lattice is expected to undergo a melting transition into an incompressible bosonic quantum Hall regime [3].

The physics of a quantum vortex lattice in bosonic superfluids attracted considerable interest in the past (for reviews see Refs. [4–7]). In a series of beautiful papers, Tkachenko laid the theoretical foundations of this field. In the incompressible limit, he demonstrated analytically that the triangular arrangement of vortices has the lowest energy [8] and determined low-energy linearly-dispersing collective excitations [9,10], known today as Tkachenko waves. In later years, the hydrodynamics of Tkachenko waves in incompressible superfluids was developed in Refs. [11–13]. With the advent of cold atom experiments, the main interest in this field shifted towards vortex lattices in compressible superfluids. These support a soft Tkachenko mode with a low-energy quadratic dispersion [14,15], whose signatures were experimentally observed in Ref. [16]. The hydrodynamics of such lattices were investigated by Baym [15,17] and later, in Ref. [18], Watanabe and Murayama proposed a low-energy effective field theory of this quantum state.[1] Finally, it is worth mentioning that a rotating superfluid in a harmonic trap maps directly to a problem of bosons in a constant magnetic field proportional to the rotation frequency.

The discrete time-reversal $T$ and parity $P$ symmetries of a two-dimensional bosonic superfluid are broken by external rotation (while its product $PT$ is preserved). In this work, we focus on consequences of the violation of these symmetries, which, to the best of our knowledge, has not been investigated before in a vortex lattice phase of a continuum superfluid.[2] Using the boson-vortex duality [24–26], we write down a low-energy effective theory of an infinite vortex lattice in a bosonic superfluid. It will be argued below that this dual formulation, where the Goldstone mode is parametrized by a gauge field, has certain advantages compared to the effective theory of Ref. [18]. After discussing the symmetries of the theory, we compute the $U(1)$ particle number and stress tensor linear responses to external sources. In addition to $P$ and $T$-invariant responses, we extract the Hall conductivity and Hall viscosity. We also investigate relations between particle number and geometric responses which follow from Galilean symmetry of the bosonic superfluid.

---

[1]An effective field theory of individual vortices was investigated recently in [19–21].

[2]We note that the Hall response was studied in a hard-core lattice model in [22, 23].

In this paper, we concentrate on the bulk properties of two-dimensional vortex lattices, and thus consider infinite uniform systems, where momentum is a good quantum number. We expect that our results should be relevant to cold atom experiments with large vortex lattices (where the angular frequency of rotation $\Omega$ approaches the transverse trapping frequency $\omega_\perp$) and numerical simulations, where periodic boundary conditions are used. Investigation of edge physics is deferred to a future work. The effective field theory developed in this paper is not applicable in the quantum Hall regime.

## 2 Dual effective theory

Boson-vortex duality [25, 26] opened an interesting perspective on the physics of two-dimensional superfluids and quantum vortices. In the dual formulation, a $U(1)$ superfluid is identified with the Coulomb phase of a two-dimensional compact $u(1)$ gauge theory without instantons [27, 28]. The dual photon has only one polarization and corresponds to the Goldstone boson of the spontaneously broken particle number symmetry. In this language, vortices are point-like charges coupled minimally to the dual $u(1)$ gauge field $a_\mu$. The latter has a finite background magnetic field fixed by the superfluid density that gives rise to the transverse Magnus force acting on vortices. In this section, we use the vortex-boson duality and formulate the low-energy effective theory of an infinite two-dimensional vortex lattice in a bosonic superfluid rotating with an angular frequency $\Omega$. In this formulation, the vortex lattice is a two-dimensional bosonic Wigner crystal—a triangular lattice of point charges embedded into a static $u(1)$-charged background that neutralizes the system, see Fig. 1. The theory is defined by the following Lagrangian

$$\mathscr{L}(e_i, b, u^i; \mathscr{A}_\mu) = \frac{m\mathbf{e}^2}{2b} - \varepsilon(b) - m\Omega b \epsilon_{ij} u^i D_t u^j + 2m\Omega e_i u^i - \mathscr{E}_{\text{el}}(u^{ij}) - \epsilon^{\mu\nu\rho} \mathscr{A}_\mu \partial_\nu a_\rho. \quad (1)$$

Here $m$ denotes the mass of the elementary Bose particle, $D_t = \partial_t + v_s^k \partial_k$ is the convective derivative, and we have introduced the dual electric and magnetic fields $e_i = \partial_t a_i - \partial_i a_t$ and $b = \epsilon^{ij} \partial_i a_j$ that are related to the coarse-grained superfluid number density $n_s$ and coarse-grained superfluid velocity $v_s^i$,

$$n_s = b, \qquad v_s^i = -\frac{\epsilon^{ij} e_j}{b}. \quad (2)$$

The first two terms in the Lagrangian (1) represent the Galilean-invariant coarse-grained superfluid characterized by the internal energy density $\varepsilon(n_s)$ (see, for example, Ref. [29]). The fields $u^i$ represent the Cartesian components of the coarse-grained displacements of the vortices from their equilibrium lattice positions. As will become explicit later, these fields are the Goldstone bosons of the translations which are spontaneously broken by the vortex lattice ground state. The third term in the Lagrangian (1) is the Magnus term that produces a force acting in the direction perpendicular to the velocity of vortices relative to the superfluid. Since the vortices are charged with respect to the dual field $a_\mu$, the term $\sim e_i u^i$ in Eq. (1) represents the dipole energy density of displaced lattice charges in the presence of a static neutralizing background. The Lagrangian also contains the elastic energy density $\mathscr{E}_{\text{el}}(u^{ij})$ of the vortex lattice which depends on the deformation tensor $u^{ij} = (\partial^i u^j + \partial^j u^i - \partial_k u^i \partial^k u^j)/2$. Its functional form is fixed by the geometry of the lattice. For a triangular vortex lattice, the elastic energy density, up to quadratic order in deformations, is [12, 15, 30]

$$\mathscr{E}_{\text{el}}^{(2)}(\partial u) = 2C_1(\partial_i u^i)^2 + C_2\big[(\partial_x u^x - \partial_y u^y)^2 + (\partial_y u^x + \partial_x u^y)^2\big], \quad (3)$$

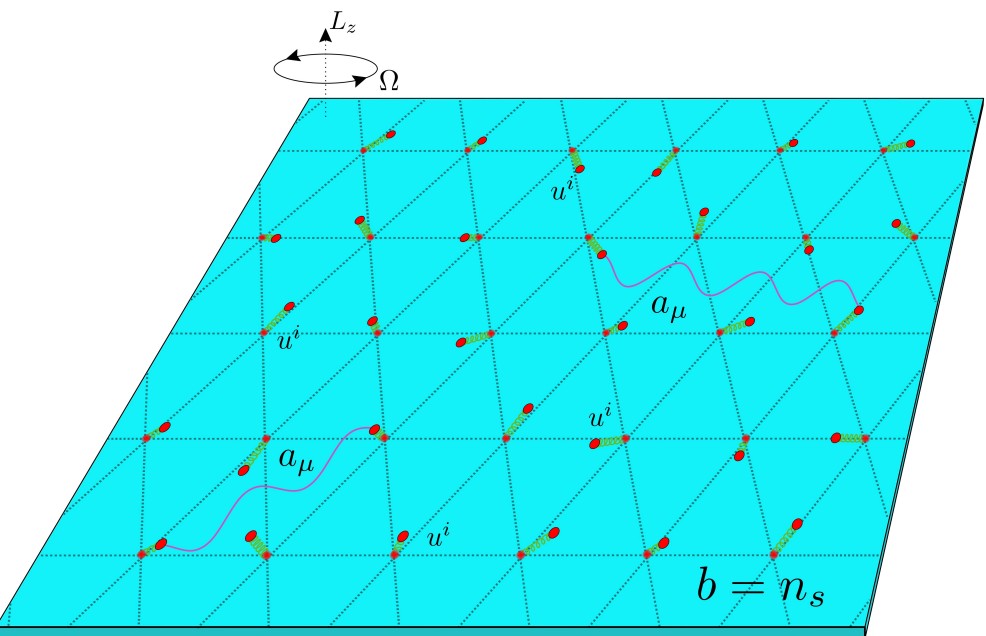

Figure 1: Dual Wigner crystal: point charges (red dots) form a triangular lattice in a homogenous neutralizing background (cyan). Microscopic displacements and photons of the dual gauge field are represented by green springs and violet wavy lines, respectively. Degrees of freedom $u^i$ and $a_\mu$ of the effective theory (1) are coarse-grained averages over large number of unit cells.

where $C_1$ and $C_2$ denote the compressional and shear modulus, respectively.[3] Notice that the bulk modulus $C_1$ does not have to be non-negative to insure the stability of the vortex lattice [12, 15]. Finally, the last term in the Lagrangian (1) takes into account the coupling of the global $U(1)$ coarse-grained current

$$j_s^\mu = -\frac{\delta S}{\delta \mathscr{A}_\mu} = \epsilon^{\mu\nu\rho}\partial_\nu a_\rho = (n_s, n_s v_s^i) \tag{4}$$

to an external $U(1)$ source field $\mathscr{A}_\mu$. Here the source is defined to vanish (up to a gauge transformation) in the ground state and thus is associated with the deviation of the external rotation frequency from its ground state value $\Omega$. For an infinite vortex lattice, the ground state is a state with $u^i = 0$, $b = n_0 = \text{const}$, $e^i = 0$, where the ground state particle density $n_0$ is fixed by the condition $d\varepsilon/db = 0$.

We emphasize that the form of the effective theory (1) is not merely a guess, but is closely related to the previous work of Watanabe and Murayama [18]. In that paper, starting from a microscopic theory of a rotating weakly-interacting Bose gas, the low-energy effective theory of the vortex lattice was derived. As we demonstrate in Appendix A, for a special choice of the energy density $\varepsilon(b)$, the Lagrangian (1) is dual to the effective theory derived in Ref. [18]. Moreover, the dual electric and magnetic fields are related to the regular part of the superfluid phase and displacement vectors via Eqs. (2) and (42). Despite being equivalent to the original theory of Ref. [18], the dual formulation (1) has an important conceptual advantage:

---

[3]The elastic properties of a two-dimensional triangular lattice are characterized by only two elastic moduli $C_1$ and $C_2$ and thus, in this respect, the lattice is indistinguishable from an isotropic medium [30]. As a result, although continuous rotation symmetry is broken spontaneously to a discrete subgroup, the theory and all observables computed in this paper respect continuous rotation symmetry. The violation of this symmetry is expected to arise from higher-derivative terms not included here.

as shown in Sec. 3, in contrast to the effective theory of Ref. [18], the linearized form of the dual theory fits naturally into a derivative expansion. This allows us to order different terms in the dual Lagrangian according to their relevance at low energies and long wave-lengths and systematically construct corrections to the leading-order theory. Later in this paper we will also construct the diffeomorphism-invariant version of the theory (1) and discuss the fate of some higher-deriviative terms not considered in [18].

Now we turn to the discussion of symmetries of the theory (1). Generically, the action of a low-energy effective theory should inherit all symmetries (irrespective of whether they are spontaneously broken and not) of the microscopic model.

First, under discrete parity and time reversal, the fields and sources transform as follows:

$$
\begin{aligned}
P &\quad x \leftrightarrow y, \quad a_t \to -a_t, \quad a_x \leftrightarrow -a_y \quad u^x \leftrightarrow u^y, \quad \mathscr{A}_x \leftrightarrow \mathscr{A}_y, \\
T &: \quad t \to -t, \quad a_t \to -a_t, \quad \mathscr{A}_i \to -\mathscr{A}_i.
\end{aligned}
\tag{5}
$$

We find that the Lagrangian (1) is not invariant separately under $P$ and $T$ since the terms proportional to $\Omega$ change sign under these transformations. The Lagrangian is invariant, however, under the combined $PT$ symmetry. Note that if one flips the sign of the rotation frequency $\Omega \to -\Omega$ under parity and time-reversal, then the theory is separately invariant under $P$ and $T$.

Second, we consider spatial translations. In a microscopic theory of a rotating Bose superfluid, the angular frequency $\Omega$ is equivalent to an effective constant magnetic field $B_{\text{eff}} = -2m\Omega$, and thus the action should be invariant under magnetic translations [18]. In an infinite vortex lattice, the ground state breaks this symmetry spontaneously. Since in the dual formulation, the fields $b$, $e_i$ and $u^i$ transform trivially under particle number $U(1)$ global symmetry, magnetic translations of the vortex lattice are implemented as usual translations on these fields. Under an infinitesimal constant spatial translation $x^i \to x^i + l^i$, the fields transform as $\delta_l \Phi = -l^k \partial_k \Phi$, where $\Phi = (e_i, b, \mathscr{A}_\mu)$, but $\delta_l u^i = -l^k \partial_k u^i + 2l^i$. As expected for a Goldstone boson of broken translations, the field $u^i$ transforms inhomogeneously. Using the Bianchi identity $\epsilon^{\mu\nu\rho} \partial_\mu \partial_\nu a_\rho = 0$, it is straightforward to check that the action $S = \int dt \, d^2x \, \mathscr{L}$ is invariant under spatial translations.

Finally, we investigate Galilean boosts. Once again, we use the fact that $b$, $e_i$ and $u^i$ are neutral under the particle number $U(1)$ symmetry, and thus an infinitesimal Galilean boost with the velocity $\beta^i$ is realized on these fields as a time-dependent spatial diffeomorphism $x^i \to x^i + \beta^i t$:

$$
\begin{aligned}
\delta_\beta b &= -\beta^k t \partial_k b, \\
\delta_\beta e_i &= -\beta^k t \partial_k e_i + b \epsilon_{ik} \beta^k, \\
\delta_\beta u^i &= -\beta^k t \partial_k u^i + 2\beta^i t.
\end{aligned}
\tag{6}
$$

On the other hand, the electric and magnetic fields constructed from the $U(1)$ source should transform as

$$
\begin{aligned}
\delta_\beta \mathscr{E}_i &= -\beta^k t \partial_k \mathscr{E}_i + \epsilon_{ij} \beta^i (\mathscr{B} - 2m\Omega), \\
\delta_\beta \mathscr{B} &= -\beta^k t \partial_k \mathscr{B},
\end{aligned}
\tag{7}
$$

where we have defined $\mathscr{E}_i = \partial_t \mathscr{A}_i - \partial_i \mathscr{A}_t$ and $\mathscr{B} = \epsilon^{ij} \partial_i \mathscr{A}_j$. The action built from the Lagrangian (1) is invariant under Galilean transformations. As we will see in the following, Galilean invariance has important consequences for the spectrum of excitations and transport properties.

# 3 Excitations and particle number transport

In this section, we work out some physical properties of the effective theory (1). In particular, we analyze its excitations and extract the $U(1)$ particle number transport coefficients such as longitudinal and Hall conductivities. To this end, it is sufficient to expand the Lagrangian (1) around the ground state $b = n_0 + \delta b$ and keep only terms quadratic in fields and sources,

$$\mathscr{L}^{(2)} = \underbrace{\frac{m}{2n_0}\mathbf{e}^2}_{\text{NLO}} \underbrace{-\frac{mc_s^2}{2n_0}\delta b^2 - n_0 m\Omega\epsilon_{ij}u^i\dot{u}^j + 2m\Omega e_i u^i - \mathscr{E}_{\text{el}}^{(2)}(\partial u) - \epsilon^{\mu\nu\rho}\mathscr{A}_\mu\partial_\nu a_\rho}_{\text{LO}}, \qquad (8)$$

where overdot denotes the time derivative and $c_s = \sqrt{n_0\varepsilon''/m}$ is the speed of sound. This Lagrangian naturally fits into a derivative expansion within the following power-counting scheme

$$a_i, u^i, \mathscr{A}_i \sim O(\epsilon^0), \qquad a_t, \partial_i, \mathscr{A}_t \sim O(\epsilon^1), \qquad \partial_t \sim O(\epsilon^2), \qquad (9)$$

where $\epsilon \ll 1$. In particular, one finds that all terms in Eq. (8), except the first one, scale as $O(\epsilon^2)$; these terms will be referred to as leading-order (LO) terms in the following. On the other hand, the electric term $\sim \mathbf{e}^2$ scales as $O(\epsilon^4)$ and thus contributes to the next-to-leading order (NLO) in this power-counting scheme. In the following, we will first work with the leading order Lagrangian and subsequently analyze the next-to-leading order corrections produced by the electric term.

## 3.1 Leading order

We first extract the excitations above the ground state from the LO part of the Lagrangian (8) in the absence of the source $\mathscr{A}_\mu$. In the LO theory Galilean symmetry is broken and the dual gauge field is not dynamical. The Gauss law $\delta S_{\text{LO}}/\delta a_t = 0$ implies $\partial_i u^i = 0$ and thus displacements are transverse. In other words, the vortex lattice is incompressible and the vortex density $n_v$ is constant in position space. The remaining four field equations are

$$c_s^2\epsilon^{ij}\partial_j\delta b + 2n_0\Omega\dot{u}^i = 0,$$

$$2m\Omega e_i - 2n_0 m\Omega\epsilon_{ij}\dot{u}^j + \partial_j\frac{\partial\mathscr{E}_{\text{el}}^{(2)}}{\partial\partial_j u^i} = 0. \qquad (10)$$

From now on, we work in the temporal gauge $a_t = 0$, where $e_i = \dot{a}_i$ and, without loss of generality, look for plane-wave solutions that propagate along the $x$ direction, i.e., where $\delta b$, $e_i$ and $u^i$ do not depend on $y$. As a result, the Gauss law now implies $u^x = 0$. In Fourier space, the field equations, written in matrix form, are

$$\begin{pmatrix} 0 & c_s^2 k^2 & -2in_0\Omega\omega \\ -i\omega & 0 & i\omega n_0 \\ 0 & -i\omega & -\frac{C_2}{m\Omega}k^2 \end{pmatrix} \begin{pmatrix} a_x \\ a_y \\ u^y \end{pmatrix} = 0. \qquad (11)$$

The linear system has a nontrivial solution only if the determinant vanishes, which fixes the dispersion relation

$$\omega = \sqrt{\frac{C_2 c_s^2}{2mn_0\Omega^2}}k^2. \qquad (12)$$

It is known that a vortex lattice in a compressible superfluid ($c_s^{-1} \neq 0$) supports the Tkachenko mode which has the dispersion (12) at small momenta [14, 15]. Moreover, since the vortex

lattice is incompressible in the LO theory, the dispersion depends only on the shear elastic modulus $C_2$, but not on the bulk modulus $C_1$. In the next subsection we will find that the inclusion of the NLO electric term gives rise to quartic corrections to the Tkachenko dispersion relation.

We now turn to the computation of the $U(1)$ particle number linear response. To this end one has to determine how the particle number current $j_s^\mu = \epsilon^{\mu\nu\rho}\partial_\nu a_\rho$ responds to variations of the $U(1)$ source $\mathscr{A}_\mu$. In particular, the density susceptibility $\chi$, the longitudinal conductivity $\sigma$ and the Hall conductivity $\sigma_H$ are defined in Fourier space as

$$\chi(\omega,k) = \frac{\delta n_s}{\delta \mathscr{A}_t}\Big|_{\omega,k},$$

$$\sigma(\omega,k) = \sigma_{xx}(\omega,k) = \frac{i}{\omega}\frac{\delta j_s^x}{\delta \mathscr{A}_x}\Big|_{\omega,k} = \frac{i}{k}\frac{\delta j_s^x}{\delta \mathscr{A}_t}\Big|_{\omega,k},$$

$$\sigma_H(\omega,k) = \sigma_{xy}(\omega,k) = -\frac{i}{\omega}\frac{\delta j_s^y}{\delta \mathscr{A}_x}\Big|_{\omega,k} = -\frac{i}{k}\frac{\delta j_s^y}{\delta \mathscr{A}_t}\Big|_{\omega,k}. \tag{13}$$

In order to extract these functions from the LO effective theory, we first solve the linearized field equations in the presence of the $U(1)$ source, substitute the solutions into the particle number current (4), and finally apply the definitions (13). As a result, we get

$$\chi(\omega,k) = \frac{C_2 k^4}{2m^2\Omega^2}\frac{1}{\omega^2 - \frac{C_2 c_s^2}{2mn_0\Omega^2}k^4},$$

$$\sigma(\omega,k) = \frac{iC_2 k^2\omega}{2m^2\Omega^2}\frac{1}{\omega^2 - \frac{C_2 c_s^2}{2mn_0\Omega^2}k^4},$$

$$\sigma_H(\omega,k) = \frac{n_0\omega^2}{2m\Omega}\frac{1}{\omega^2 - \frac{C_2 c_s^2}{2mn_0\Omega^2}k^4}. \tag{14}$$

In the static regime $\omega = 0$, we find $\chi(k) = -\frac{n_0}{mc_s^2}$, which satisfies the compressibility sum rule $\chi(k=0) = -\partial n/\partial \mu = -\frac{n_0}{mc_s^2}$. We observe that the gapless Tkachenko excitation saturates the transport of particle number at low energies and long wavelengths.

## 3.2 Beyond the leading order

We now go beyond the LO. We will not try to construct the most general NLO Lagrangian, but only include the NLO electric term, which has important physical consequences. First, it will become manifest later that the Galilean symmetry, lost at leading order, is now restored. Second, the Gauss law now reads

$$\partial_i\big(u^i + \frac{1}{2n_0\Omega}e^i\big) = 0, \tag{15}$$

and thus the vortex lattice becomes compressible and the displacement field $u^i$ is not transverse anymore.

The calculation of the dispersion of excitations is straightforward, but tedious; here we present only the main results, see also Fig. 2. In the presence of the electric term one finds two physical modes. The first mode is the Tkachenko mode, which is now elliptically polarized[4]

$$\frac{u^x}{u^y} = i\sqrt{\frac{C_2 c_s^2}{8mn_0\Omega^4}k^2} + O(k^4), \tag{16}$$

---

[4] As before, in this subsection we consider a plane-wave ansatz with momentum $\mathbf{k} = (k,0)$.

and has the dispersion

$$\omega = \sqrt{\frac{C_2 c_s^2}{2mn_0\Omega^2}\left[k^2 - \frac{2C_2 + mn_0 c_s^2}{8mn_0\Omega^2}k^4 + O(k^6)\right]}. \tag{17}$$

In addition, one finds the gapped Kohn mode with the dispersion

$$\omega = 2|\Omega|\left[1 + \frac{4(C_1 + C_2) + mn_0 c_s^2}{8mn_0\Omega^2}k^2 + O\left(k^4\right)\right]. \tag{18}$$

At zero momentum this mode is circularly polarized. We observe that the Galilean symmetry of the problem is restored by the NLO electric term and ensures that the high-energy Kohn mode is properly captured by the low-energy effective theory.

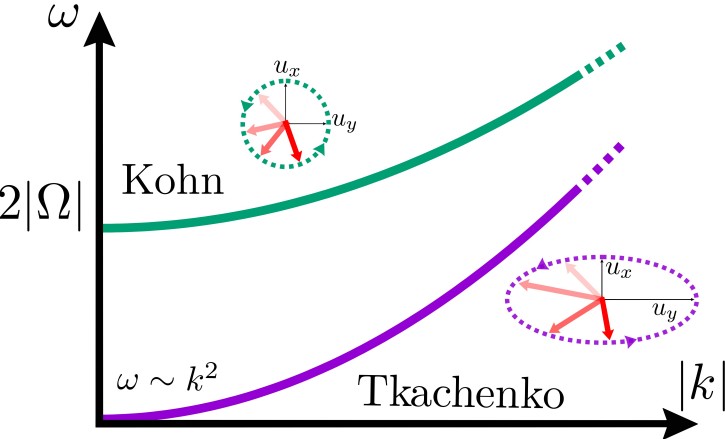

Figure 2: Sketch of dispersions and polarizations of Tkachenko and Kohn excitations.

The computation of the particle number linear response follows the same steps as described in the Sec. 3.1. The analytical expressions for $\chi$, $\sigma$ and $\sigma_\mathrm{H}$ are cumbersome. For this reason, here we limit our discussion of the $U(1)$ response functions to a few special regimes.

We start with the density susceptibility $\chi$ which vanishes in the homogeneous case $k = 0$, $\omega \neq 0$. This makes sense since particle density should not change under variations of a uniform time-dependent electrostatic potential. In the static regime, the compressibility sum rule $\chi(k \to 0) = -\partial n/\partial \mu = -\frac{n_0}{mc_s^2}$ is satisfied.

Now we turn to the conductivities. In the static regime $\omega = 0$, we find that the vortex lattice behaves as an insulator, i.e., $\sigma(k) = \sigma_\mathrm{H}(k) = 0$. Consider now the regime of finite $\omega$, but small $k$. Expanding conductivities in momentum around $k = 0$, one finds[5]

$$\sigma(\omega, k) = i\frac{n_0\omega}{m(\omega^2 - 4\Omega^2)} + i\frac{mn_0\omega^2 c_s^2 + 2C_2\left(\omega^2 + 4\Omega^2\right) + 4C_1\omega^2}{m^2\omega\left(\omega^2 - 4\Omega^2\right)^2}k^2 + O\left(k^4\right),$$

$$\sigma_\mathrm{H}(\omega, k) = -\frac{2n_0\Omega}{m(\omega^2 - 4\Omega^2)} - \frac{2\Omega\left(mn_0 c_s^2 + 4(C_1 + C_2)\right)}{m^2\left(\omega^2 - 4\Omega^2\right)^2}k^2 + O\left(k^4\right). \tag{19}$$

---

[5] In general, the conductivities depend on the frequency $\omega$ and the momentum vector $\mathbf{k}$. In the light of our ansatz, the expressions (19) are only valid for momenta $\mathbf{k} = (k, 0)$. Generalization of this result to arbitrary, but small momentum $\mathbf{k}$ will be found in Sec. 7.

The first terms in the Taylor expansion are exact conductivities in the homogeneous $k = 0$ regime and their form is fully fixed by the Kohn theorem. In Sec. 7 it will turn out to be convenient to combine the longitudinal and Hall conductivities into the leading order conductivity tensor

$$\sigma_{ij}^{(0)}(\omega) = \frac{n_0}{m(\omega^2 - 4\Omega^2)} \big( i\omega\delta_{ij} - 2\Omega\epsilon_{ij} \big). \tag{20}$$

We will show in Sec. 7 that the finite-momentum quadratic corrections in Eq. (19) are tied to the geometric response very much in the spirit of Refs. [31, 32].

Formally, it is possible to extract the leading order result (14) from the response functions discussed here. To this end, we introduce a small parameter $\delta$ and replace $\omega \to \delta^2\omega$ and $k \to \delta k$ in response functions. The leading order of the Taylor expansion in $\delta$ of the functions $\chi$, $\sigma$ and $\sigma_\mathrm{H}$ gives exactly (14).

Finally, it is important to remark again that, in this paper, we do not attempt to construct the most general theory that includes all NLO terms that are consistent with symmetries. As a result, the subleading corrections to observables [such as the quartic term in the Tkachenko dispersion (17) and the quadratic terms in conductivities (19)] might be modified by the omitted NLO terms. A systematic investigation of the most general NLO theory is postponed to a future study.

# 4 Difeomorphism-invariant formulation of the effective theory

One might be not fully satisfied with the effective theory (1) for the following reason: Although the displacement field $u^i$ carries a spatial index, it does not transform as a vector field under spatial general coordinate transformations (diffeomorphisms) because it is the Goldstone mode of spontaneously broken magnetic translations. Hence, the generalization of the theory (1) to a form valid in general curvilinear coordinate is not straightforward. In order to circumvent this problem, we introduce here an alternative formalism used previously to describe solids [33–35]. Instead of displacements, we introduce a set of scalar fields $X^a(t, \mathbf{x})$, with $a = 1, 2$, that represent the Lagrange coordinates frozen into the vortex lattice. In other words, any vortex has a constant coordinate $X^a$ along its worldline. Imagine now a two-dimensional curved surface parametrized by a general set of spatial coordinates $x^i$ with a geometry given by a metric tensor $g_{ij}$. In these coordinates, the effective action of the vortex lattice is given by $S = \int dt\, d^2x\, \sqrt{g}\, \mathscr{L}$, with the scalar Lagrangian

$$\mathscr{L} = \frac{m g^{ij} e_i e_j}{2b} - \varepsilon(b) - \pi n_v \varepsilon^{\mu\nu\rho} \epsilon_{ab} a_\mu \partial_\nu X^a \partial_\rho X^b - \mathscr{E}_\mathrm{el}(U^{ab}) - \varepsilon^{\mu\nu\rho} A_\mu \partial_\nu a_\rho, \tag{21}$$

where $g = \det g_{ij}$, $b = \epsilon^{ij}\partial_i a_j/\sqrt{g}$, $\varepsilon^{\mu\nu\rho} = \epsilon^{\mu\nu\rho}/\sqrt{g}$ and $U^{ab} = g^{ij}\partial_i X^a \partial_j X^b$. The vortex number current $j_v^\mu \sim \varepsilon^{\mu\nu\rho} \epsilon_{ab} \partial_\nu X^a \partial_\rho X^b$ couples to the dual gauge field $a_\mu$. In contrast to the theory introduced in Sec. 2, in this formulation, the $U(1)$ source $A_\mu$ has a finite background magnetic field $B = \epsilon^{ij}\partial_i A_j = -2m\Omega$. There is no unique way how the Lagrange coordinates are defined in a solid, which leads to global symmetries that act in internal space. In particular, the action must be invariant under constant internal shifts $X^a \to X^a + l^a$. In addition, the theory is also invariant under discrete internal rotations that map the triangular lattice to itself. This symmetry constraints the form of the elastic term $\mathscr{E}_\mathrm{el}(U^{ab})$. With $n_v$ transforming as a scalar, the action is invariant under spatial general coordinate transformations and is thus an ideal starting point for the computation of geometric responses.

The non-linear theory (21) fits naturally into a derivative expansion with the following power-counting scheme ($\epsilon \ll 1$)

$$a_i, X^a, A_i \sim O(\epsilon^{-1}), \qquad a_t, A_t \sim O(\epsilon^0), \qquad \partial_i \sim O(\epsilon^1), \qquad \partial_t \sim O(\epsilon^2). \tag{22}$$

The difference in the scaling of space and time originates from the quadratic dispersion of the soft Tkachenko mode. In this power counting, the first term in the Lagrangian (21) is of order $O(\epsilon^2)$ and becomes the next-to-leading order correction to the remaining terms in Eq. (21) that all scale as $O(\epsilon^0)$ and thus constitute the leading-order part. In Appendix B, we demonstrate that in Cartesian coordinates of flat space, where $u^i = x^i - \delta_a^i X^a$, the Lagrangian (21) reduces to the original theory (1). In that case, in the ground state $n_\nu = -B/(2\pi) = m\Omega/\pi$ and thus $n_\nu$ represents the ground state number density of vortices in flat space.

The Maxwell equations that follow from the Lagrangian (21) are

$$\tilde{B} + \pi n_\nu \varepsilon^{ij} \epsilon_{ab} \partial_i X^a \partial_j X^b = 0, \tag{23}$$

$$\tilde{E}_j + 2\pi n_\nu \epsilon_{ab} \dot{X}^a \partial_j X^b - \varepsilon''(b)\partial_j b = 0, \tag{24}$$

where we have introduced $\tilde{B} = B - m\varepsilon_j^i \partial_i v_s^j$ and $\tilde{E}_j = E_j - m(\dot{v}_{sj} + g_{mn} v_s^m \partial_j v_s^n)$. By taking the variation of the action with respect to $X^a$ we find

$$\pi n_\nu \varepsilon^{\mu\nu\rho} \epsilon_{ab} \partial_\mu a_\nu \partial_\rho X^b - \frac{1}{\sqrt{g}} \partial_j \left( \sqrt{g} \frac{\partial \mathscr{E}_{\mathrm{el}}}{\partial U^{ab}} g^{ij} \partial_j X^b \right) = 0. \tag{25}$$

## 5 Stress tensor and geometric response

In this section, we extract from the Lagrangian (21) the stress tensor and evaluate its linear response to an external metric perturbation. Our main aim here is to compute the viscosity tensor $\eta^{ijkl}$ which can be extracted from the linear response formula $\delta T^{ij} = -\lambda^{ijkl}\delta g_{kl} - \eta^{ijkl}\delta\dot{g}_{kl}$.

First, following Refs. [33, 34], we express the elastic energy density as

$$\mathscr{E}_{\mathrm{el}}(U^{ab}) = \sqrt{U}\varepsilon_{\mathrm{el}}(U^{ab}), \tag{26}$$

where $U = \det U^{ab}$. In this parametrization, the ground state is fixed by the expression $\partial \varepsilon_{\mathrm{el}}/\partial U^{ab} = 0$. It is straightforward now to compute the stress tensor

$$T^{ij} = \frac{2}{\sqrt{g}} \frac{\delta S}{\delta g_{ij}} = \underbrace{P g^{ij} + \rho v_s^i v_s^j}_{T_{\mathrm{ideal}}^{ij}} \underbrace{-2\sqrt{U} \frac{\partial \varepsilon_{\mathrm{el}}}{\partial U^{ab}} \partial^i X^a \partial^j X^b}_{T_{\mathrm{el}}^{ij}}, \tag{27}$$

where its ideal part comes from the superfluid terms in the action, while the elastic part originates from the elastic energy. Notice that the Magnus term (the third term in the Lagrangian (21)) is topological and does not contribute to the stress tensor.

Consider now the linear response of the stress tensor to a metric source $g_{ij} = \delta_{ij} + h_{ij}$. First, we have to linearize the equations of motion (23), (24), (25). We write $X^a = \delta_i^a x^i - u^a$ and $b = b_0 + \delta b$ and get

$$\frac{1}{b_0} \partial_i e^i + 2\Omega \partial_i u^i = 0,$$

$$\frac{m}{b_0} \epsilon_{jk} \dot{e}^k + 2m\Omega \epsilon_{ja} \dot{u}^a - \varepsilon''(b_0) \partial_j \delta b = 0,$$

$$-m\Omega \epsilon_{ab}(b_0 \dot{u}^b + \epsilon^{ib} e_i) + \frac{\partial \mathscr{E}_{\mathrm{el}}}{\partial U^{ab}} \left( \Delta u^b - \frac{1}{2} \delta^{ij} \partial^b h_{ij} + \delta^{bc} \partial^i h_{ic} \right) - \frac{\partial^2 \mathscr{E}_{\mathrm{el}}}{\partial U^{ab} \partial U^{cd}} \partial^b U^{cd} = 0, \tag{28}$$

where $h = h^i_{\ i}$ and

$$\frac{\partial \mathscr{E}_{\mathrm{el}}}{\partial U^{ab}} = \frac{\partial}{\partial U^{ab}}\Big[\sqrt{U}\varepsilon_{\mathrm{el}}(U^{ab})\Big] = \frac{\varepsilon_{\mathrm{el}}}{2\sqrt{U}}\frac{\partial U}{\partial U^{ab}} + \sqrt{U}\frac{\partial \varepsilon_{\mathrm{el}}}{\partial U^{ab}}. \tag{29}$$

In the homogeneous regime ($k = 0$), we find that that all $h_{ij}$-dependent terms drop out from the linearized equations of motions. As a result, the on-shell stress tensor does not depend on time derivatives of the metric source $h_{ij}$ and thus the AC viscosity tensor $\eta^{ijkl}(\omega)$ vanishes trivially in our theory. If in addition one assumes that in the ground state $\varepsilon_{\mathrm{el}} = 0$, the expression (29) vanishes resulting in a stronger result $\eta^{ijkl}(\omega, \mathbf{k}) = 0$.

The absence of the bulk and shear viscosity coefficients is completely expected since an effective theory defined by a real action cannot dissipate energy at zero temperature. It is well-known however that two-dimensional systems with broken time-reversal and parity symmetries (such as quantum Hall fluids, chiral superfluids, etc) generically exhibit a non-dissipative viscous Hall response [36–39]. Notwithstanding, we found here that the effective theory defined by the Lagrangian (21) has zero Hall viscosity. Since the theory (21) might be incomplete at the next-to-leading order, it is natural to wonder if the Hall viscosity actually vanishes in the vortex lattice phase of a bosonic superfluid. In the next section we provide some arguments in favor of that.

# 6 Hall viscosity and coupling to spin connection

In effective theories of quantum fluids the coupling of currents to the spin connection is as a rule quantized and gives rise to a finite Hall viscosity at zero frequency and momentum [40,41]. The spin connection $\omega_\mu$ is built from the orthonormal spatial vielbein $e^a_i$ as follows

$$\omega_t = \frac{1}{2}\epsilon^{ab}e^{aj}\partial_t e^b_j, \qquad \omega_i = \frac{1}{2}\epsilon^{ab}e^{aj}\nabla_i e^b_j. \tag{30}$$

It transforms as an abelian gauge field under local rotations in the internal vielbein space (indices $a, b = 1, 2$). The magnetic field constructed from this gauge field is proportional to the Ricci curvature of the two-dimensional surface

$$B_\omega = \varepsilon^{ij}\partial_i \omega_j = \frac{1}{2}R. \tag{31}$$

In our problem there could be NLO terms (omitted above) that couple particle or vortex currents to the spin connection

$$\mathscr{L}_\omega = s\omega_\mu j^\mu_s + s_v \omega_\mu j^\mu_v, \tag{32}$$

where $s, s_v$ are constant coefficients. For a finite density of particles $j^t_s = n_s$ or vortices $j^t_v = n_v$, Eq. (32) introduces in the effective action a term that is linear in $\omega_t$ that generates a finite Hall viscosity. In this section we determine the values of $s$ and $s_v$.

In chiral fermionic superfluids $P$ and $T$ are spontaneously broken and there (in the absence of a vortex lattice) the coupling (32) completely determines the Hall viscosity. This was analyzed in detail in [29,42]. Note however that in a non-rotating bosonic superfluid (and also in a fermionic $s$-wave superfluid) the coupling to the particle current is forbidden by time reversal invariance, thus $s = 0$. A physical interpretation of this is that elementary bosons in a bosonic superfluid (Cooper pairs in s-wave superfluids) do not have internal spin. This should not change in the vortex lattice phase, and thus we can set $s = 0$.

The term that couples the vortex current to the spin connection, on the other hand, is both $P$ and $T$ invariant. In principle it could be non-vanishing in the present problem and would give rise to a non-zero Hall viscosity of the vortex lattice. We expect that this term

fully determines the Hall viscosity, but a full analysis of all NLO terms would be necessary to be completely certain. In addition, if $s_v$ is non-zero, the Magnus force acting on a vortex in curved space is modified from the flat space result [43, 44] by a term that is proportional to the spatial curvature (see Appendix C).

Even though it is not forbidden by symmetries, in this section we provide arguments that in a spinless bosonic superfluid the vortex current does not couple to the spin connection and thus we expect the Hall viscosity of the vortex lattice to be zero. To see this, one has to compute the Berry phase accumulated by a single vortex traversing a closed loop in parameter space on a spatial surface where a finite background of the spin connection source is present. Simple examples of such processes are (i) a static vortex in flat space under periodic homogeneous shear deformations which give rise to a finite temporal component of the spin connection, (ii) a vortex traversing a closed loop on a time-independent curved surface (such as a sphere).

We start from the Gross-Pitaevskii mean-field theory, where the Berry phase accumulated by a vortex over a closed loop in parameter space is given by the action

$$S_{\text{Berry}} = i \int_0^T dt \int d\mathbf{x} \sqrt{g} \psi^\dagger \overleftrightarrow{D}_t \psi = -\int_0^T dt \int d\mathbf{x} \sqrt{g} n_s D_t \phi, \tag{33}$$

where the order parameter is $\psi = \sqrt{n_s} e^{i\phi}$, the convective derivative is $D_t = \partial_t + V^i \partial_i$ and $\overleftrightarrow{D}_t = (\overrightarrow{D}_t - \overleftarrow{D}_t)/2$. Here $V^i$ is a regular background velocity field which can be found by removing the contribution from the vortex defect. The Berry phase defined above is general coordinate invariant and thus one can work in any coordinate system to compute it. Now we can rewrite the Berry phase (33) in the dual language. Using the relations $n_s = \varepsilon^{ij} \partial_i a_j$, $j_s^i = -\varepsilon^{ij}(\partial_t a_j - \partial_j a_t)$ and the definition of the vortex current $j_v^\mu = \frac{1}{2\pi} \varepsilon^{\mu\nu\rho} \partial_\nu \partial_\rho \phi$ it is straightforward to find

$$S_{\text{Berry}} = -2\pi \int_0^T dt \int d\mathbf{x} \sqrt{g} a_\mu j_v^\mu - \text{self-energy dynamical contribution}. \tag{34}$$

The self-energy subtraction is needed because this term generates the dynamical part of the phase (which is proportional to the time $T$) and thus does not contribute to the Berry phase. Since Eq. (33) is the only term in the Gross-Pitaevskii functional that contributes to the Berry phase, we have just demonstrated that vortex defects of a bosonic superfluid in the mean-field theory couple only to the gauge field $a_\mu$, but not to the spin connection. The Magnus force calculation is consistent with this result (see Appendix C). Notice however that the Gross-Pitaevskii theory only takes into account the macroscopically occupied condensate and misses corrections originating from microscopically occupied Bogoliubov quasiparticles. This implies that the above argument only rules out the contribution to the coupling $s_v$ which scales as the total number of particles $N$.

In order to compute the Berry phase with accuracy of order unity in the particle number one has to go beyond the Gross-Pitaevskii approximation and include Gaussian fluctuations around the mean-field vortex state. This results in the Bogoliubov-corrected ground state (vacuum of Bogoliubov quasiparticles) instead of just the coherent mean-field ground state. This approximation was used in Ref. [45] to compute the Berry phase of a vortex[6] traversing a closed loop in a Bose superfluid defined on a sphere. For an infinitesimal loop, the Berry phase was found to be proportional to the total number $N$ of bosons on the sphere times the solid angle swept by the loop. The Berry phase on a sphere is thus in essence identical to the Berry

---

[6]To be precise, since only one elementary vortex cannot be defined on a sphere, the authors of Ref. [45] considered an antipodal vortex-antivortex pair configuration. They calculated the Berry phase collected by the vortex and antivortex that traverse two small loops close to the poles. Every loop contributes the same amount to the total Berry phase.

phase of a vortex moving on a plane [43]. The absence of a term in the Berry phase of order unity (i.e. independent of $N$) that is proportional to the curvature of the sphere thus suggests that vortices do not couple to the spin connection in a bosonic superfluid. This implies that in a bosonic superfluid vortices do not carry internal spin.

There is one possible loophole to the argument presented above. What we have just computed is the coupling of a single vortex to the spin connection in an effective theory where the coordinate of the vortex is a degree of freedom. On the other hand, Eq. (32) is written for a theory where the individual vortices have been smoothed over so that the degrees of freedom are now the fields $X^a$. Whether coupling to spin connection appears or not during this transition from one description to another is, strictly speaking, an open question.

# 7 Relations between elasticity, viscosity and conductivity

Galilean invariance gives rise to remarkable relations between particle number and geometric responses. In quantum fluids these relations were put forward in Refs. [31, 32]. Here we investigate these relations in the context of a vortex lattice in a Galilean-invariant bosonic superfluid.

The relations that we want to discuss here are valid in flat space and can be obtained as follows. First, one expands the conductivity tensor $\sigma_{ij}(\omega, \mathbf{k})$ in a Taylor series in momentum

$$\sigma_{ij}(\omega, \mathbf{k}) = \sigma_{ij}^{(0)}(\omega) + \sigma_{ij}^{(2)}(\omega, \mathbf{k}) + \dots. \tag{35}$$

It was shown in [32] that Galilean invariance implies the following relation

$$\sigma_{in}^{(2)}(\omega, \mathbf{k}) = -\sigma_{ij}^{(0)}(\omega) \frac{1}{n_0} k_k \chi^{kjlm}(\omega) \frac{1}{n_0} k_l \sigma_{mn}^{(0)}, \tag{36}$$

where $n_0$ is the particle number density and the tensor $\chi^{kjlm}$ is given by

$$\chi^{kjlm}(\omega) = \frac{1}{i\omega} \frac{\partial T^{kj}}{\partial u_{lm}} + \eta^{kjlm}. \tag{37}$$

In the case of fluids [32], the first term in Eq. (37) reduces to $i\kappa^{-1}\delta^{kj}\delta^{lm}/\omega$, where $\kappa^{-1} = -V(\partial P/\partial V)$ is the inverse compressibility. In our problem the stress tensor contains also the elastic part

$$T^{ij} = P\delta^{ij} + \rho v_s^i v_s^j - 4C_1 \delta^{ij} u_{kk} - 2C_2(\delta^{ik}\delta^{jl} + \delta^{jk}\delta^{il} - \delta^{ij}\delta^{kl})u_{kl}, \tag{38}$$

which substituted into Eq. (37) gives

$$\chi^{kjlm}(\omega) = \frac{i}{\omega}\Big( \underbrace{[mn_0 c_s^2 + 4C_1]}_{\kappa^{-1}}\delta^{kj}\delta^{lm} + 2C_2[\delta^{kl}\delta^{jm} + \delta^{jl}\delta^{km} - \delta^{kj}\delta^{lm}]\Big) + \eta^{kjlm}. \tag{39}$$

Putting now this result into Eq. (36) and using $\eta^{kjlm} = 0$ and Eq. (20), it is straightforward to check that the quadratic terms in conductivities (19) satisfy the relation (36) for $\mathbf{k} = (k, 0)$. This calculation confirms the validity of Eq. (36) in quantum solids.

# 8 Discussion and outlook

In this paper we constructed an effective theory of a quantum vortex lattice in a bosonic Galilean-invariant compressible superfluid. We note that our theory (21) does not have the

most general form consistent with symmetries. Even at leading order, based only on symmetries, the energy $\mathscr{E}$ could be any function of the dual magnetic field $b$, the strain $U^{ab}$ and the background magnetic field $\tilde{B}$ that was introduced in Sec. 4. This function does not need to have the form of the sum $\varepsilon(b) + \mathscr{E}_{\mathrm{el}}(U^{ab})$ as was assumed in Eq. (21). At next-to-leading order we analyzed the fate of some terms, but did not construct all possible terms allowed by symmetry. Despite these shortcomings, we believe that our theory captures properly the excitations and linear response of the quantum vortex lattice. In the future it would be important to perform a systematic construction of the effective theory in its most general form.

Since the parity and time-reversal symmetries are broken in the vortex lattice phase, the Hall viscosity is not prohibited by symmetries. Moreover, the Hall viscosity was found to be nontrivial in a somewhat related problem of chiral vortex fluids [46, 47]. Nevertheless, the effective theory analyzed in this paper gave rise to a vanishing Hall viscosity at zero frequency and momentum. As we discussed, neither particles nor vortices couple to the spin connection, so we expect the Hall viscosity to be zero even though we cannot make a definitive statement as we did not analyze all NLO corrections in the effective theory. A systematic NLO construction is deferred to a future work.

In addition to the Hall viscosity, time-reversal breaking crystals exhibit an independent viscoelastic response known as the phonon Hall viscosity [48]. In contrast to the Hall viscosity, which quantifies the response of the stress tensor to a time-dependent background metric, the phonon Hall viscosity fixes the response to a time-dependent strain.[7] In this paper we did not attempt to extract the phonon Hall viscosity and it is an open problem for the future.

Regular vortex lattices were also observed in cold atom experiments with rotating fermionic s-wave superfluids [49]. It would be interesting to apply the effective theory of this paper to these systems. Moreover, vortex lattices should also be formed in rotating chiral superfluids and it would be interesting to construct effective theories of these states and apply these theories to rotating $^3$He-A superfluids.

The physics of vortices on curved surfaces is fascinating, for review see e.g. [50]. It would be very interesting to apply our effective theory to vortex lattices that live on curved substrates.

Finally, one may wonder if the effective theory developed here can be directly applied to a thin superconducting film in an external perpendicular magnetic field. It is known that in this systems in the absence of disorder the triangular vortex lattice is stable under perturbations [51] and is a good candidate for the ground state. In addition, due to inefficient screening the vortices interact logarithmically [52] up to the Pearl length $\Lambda = 2\lambda_L^2/d$, where $\lambda_L$ is the London penetration length and $d$ is the width of the film. For thin films ($\lambda_L \gg d$) the Pearl length can be very large. Nevertheless, it was shown in [51] that the dispersion relation of lattice vibrations scale at low momenta as $\omega \sim k^{3/2}$ which differs from the quadratic Tkachenko dispersion. Fractional dispersion at low momenta originates from the coupling to the electromagnetic field that propagates in three spatial dimensions. We thus expect that our effective theory of the vortex lattice can be employed also in clean thin superconducting films after dynamical electromagnetism is included.

*Acknowledgments* – It is our pleasure to acknowledge discussions with Sasha Abanov, Assa Auerbach, Iacopo Carusotto, Matt Roberts, Haruki Watanabe and Wilhelm Zwerger. Our work is supported by the Emmy Noether Programme of German Research Foundation (DFG) under grant No. MO 3013/1-1. C. H. is supported by the Ramon y Cajal fellowship RYC-2012-10370, the Spanish national grant MINECO-16-FPA2015-63667-P and the Asturian grant FC-15-GRUPIN14-108. D. T. S. is supported, in part, U.S. DOE Grant No. DE-FG02-13ER41958, the ARO MURI Grant No. 63834-PH-MUR, and a Simons Investigator Grant from the Simons Foundation.

---

[7]For the purpose of the calculation of the phonon Hall viscosity, the phonon quantum fluctuation are assumed to be frozen and the strain is treated as an external source, not a dynamical field.

## A Relation to the effective theory of Watanabe and Murayama

In Ref. [18] Watanabe and Murayama started from the microscopic Lagrangian of a two-dimensional weakly-coupled repulsive Bose gas that rotates with the angular frequency $\Omega$ and is trapped in a harmonic potential of frequency $\omega$ which is larger but very close to $\Omega$

$$\mathscr{L} = i\psi^\dagger \overset{\leftrightarrow}{\partial_t}\psi - \frac{1}{2m}\left|(\partial_i - im\Omega\epsilon_{ij}x^j)\psi\right|^2 - V_{\text{eff}}(x)\psi^\dagger\psi - \frac{1}{2}g(\psi^\dagger\psi)^2, \tag{40}$$

where $\overset{\leftrightarrow}{\partial_t} = (\overset{\rightarrow}{\partial_t} - \overset{\leftarrow}{\partial_t})/2$ and $V_{\text{eff}}(x) = m(\omega^2 - \Omega^2)x^2/2$. In a series of steps they arrived to a low-energy non-linear effective theory of an (essentially) infinite vortex lattice. In the presence of the $U(1)$ source $\mathscr{A}_\mu$ their theory is encoded in the Lagrangian

$$\mathscr{L}_{\text{WM}} = \frac{1}{g}\left(\mu_0 - \dot\varphi - \mathscr{A}_t - m\Omega\epsilon_{ij}u^i\dot u^j - \frac{1}{2m}\left[\partial_i\varphi + \mathscr{A}_i + 2m\Omega\epsilon_{ij}u^j + m\Omega\epsilon_{kl}u^k\partial_i u^l\right]^2\right)^2 - \mathscr{E}_{\text{el}}(u^{ij}), \tag{41}$$

$\mu_0$ is the chemical potential and $\varphi$ is the regular part of the superfluid phase. The superfluid density $n_s$ and the current $j_s^i$ are easy to compute

$$n_s = -\frac{\delta S_{\text{WM}}}{\delta\mathscr{A}_t} = \frac{2}{g}\left(\mu_0 - \dot\varphi - \mathscr{A}_t - m\Omega\epsilon_{ij}u^i\dot u^j - \frac{1}{2m}\left[\partial_i\varphi + \mathscr{A}_i + 2m\Omega\epsilon_{ij}u^j + m\Omega\epsilon_{kl}u^k\partial_i u^l\right]^2\right),$$

$$j_s^i = -\frac{\delta S_{\text{WM}}}{\delta\mathscr{A}_i} = \frac{n_s}{m}\delta^{ij}\left(\partial_j\varphi + \mathscr{A}_j + 2m\Omega\epsilon_{jk}u^k + m\Omega\epsilon_{kl}u^k\partial_j u^l\right). \tag{42}$$

In this Appendix we demonstrate that for a special choice of the internal energy $\varepsilon(b)$ the Lagrangian (1) is dual to the Lagrangian $\mathscr{L}_{\text{WM}}$. The two theories are related by the Legendre transformation

$$\mathscr{L}(b, e_i) = \mathscr{L}_{\text{WM}}(\dot\varphi, \partial_i\varphi) + \dot\varphi b - \epsilon^{ij}\partial_i\varphi e_j. \tag{43}$$

Using now $n_s = b$ and $j_s^i = -\epsilon^{ij}e_j$ in Eq. (42) we find

$$\partial_i\varphi = -m\frac{\epsilon_{ij}e^j}{b} - \mathscr{A}_i - 2m\Omega\epsilon_{ij}u^j - m\Omega\epsilon_{kl}u^k\partial_i u^l,$$

$$\dot\varphi = \mu_0 - \frac{gb}{2} - \mathscr{A}_t - m\Omega\epsilon_{ij}u^i\dot u^j - \frac{me^2}{2b^2}. \tag{44}$$

With the help of these expressions we can eliminate now the derivatives of the phase $\varphi$ from the right-hand-side of Eq. (43). As a result, we arrive at the Lagrangian (1) with the energy density $\varepsilon(b) = gb^2/2 - \mu_0 b$.

## B Equivalence of the Lagrangians (1) and (21) in Cartesian coordinates

In this appendix we demonstrate that the diffeomorphism-invariant theory defined by the Lagrangian (21) reduces in Cartesian coordinates to (1). In this case, $g_{ij} = \delta_{ij}$ and Eq. (21) simplifies to

$$\mathscr{L} = \frac{m\delta^{ij}e_i e_j}{2b} - \varepsilon(b) - \pi n_v\epsilon^{\mu\nu\rho}\epsilon_{ab}a_\mu\partial_\nu X^a\partial_\rho X^b - \mathscr{E}_{\text{el}}(U^{ab}) - \epsilon^{\mu\nu\rho}A_\mu\partial_\nu a_\rho. \tag{45}$$

In addition, in these coordinates we can choose $X^a = \delta_i^a (x^i - u^i)$ which implies

$$-\pi n_v \epsilon^{\mu\nu\rho} \epsilon_{ab} a_\mu \partial_\nu X^a \partial_\rho X^b \rightarrow -m\Omega b \epsilon_{ij} u^i D_t u^j + 2m\Omega e_i u^i - 2m\Omega a_t, \tag{46}$$

where we dropped surface terms and used $n_v = m\Omega/\pi$. Now the last term in Eq. (46) is compensated by the contribution from the last term in Eq. (45) since the source $A_\mu$ has a finite background magnetic field $B = -2m\Omega$. This results in a simple shift of the source $A_\mu \rightarrow \mathscr{A}_\mu$ which now has zero background magnetic field. Finally, in Cartesian coordinates

$$U^{ab} = \delta^{ij} \partial_i X^a \partial_j X^b = \delta^{ab} - \underbrace{\left( \partial^a u^b + \partial^b u^a - \partial^c u^a \partial^c u^b \right)}_{2u^{ab}}, \tag{47}$$

and thus $U^{ab}$ is fully determined by the deformation tensor $u^{ab}$.

## C  Coupling to spin connection and Magnus force

Let us consider the terms in the effective action that couple the vortex current to the gauge field $a_\mu$ and the spin connection $\omega_\mu$

$$S = -\int dt\, d^2x \, \sqrt{g} \left( q_v a_\mu + s_v \omega_\mu \right) j_v^\mu, \tag{48}$$

where $q_v$ and $s_v$ are constant charges. Consider now the current produced by a point-like vortex

$$j_v^\mu = \frac{1}{\sqrt{g}} \int d\tau \, \dot{x}_v^\mu \delta^{(3)}(x - x_v(\tau)), \tag{49}$$

where $x_v^\mu(\tau) = (\tau, \mathbf{x}_v(\tau))$. Then, the action is

$$S = -\int d\tau \, \dot{x}_v^\mu \left( q_v a_\mu(x_v) + s_v \omega_\mu(x_v) \right). \tag{50}$$

We can compute the force from the variation of the action with respect to the position of the vortex

$$
\begin{aligned}
F_i = \frac{\delta S}{\delta x_v^i} &= -\left[ \dot{x}_v^\mu \left( q_v \partial_i a_\mu + s_v \partial_i \omega_\mu \right) - \frac{d}{d\tau} \left( q_v a_i(x_v) + s_v \omega_i(x_v) \right) \right] \\
&= -\dot{x}_v^\mu \left( q_v (\partial_i a_\mu - \partial_\mu a_i) + s_v (\partial_i \omega_\mu - \partial_\mu \omega_i) \right) \\
&= (q_v e_i + s_v E_{\omega i}) - \varepsilon_{ij} v_v^j (q_v b + s_v B_\omega),
\end{aligned}
\tag{51}
$$

where $v_v = dx_v/d\tau$, $E_{\omega i} = \partial_t \omega_i - \partial_i \omega_t$ and $B_\omega = \varepsilon^{ij} \partial_i \omega_j$. Using now the relation to the superfluid density and current

$$b = n_s, \quad e_i = \varepsilon_{ij} j_s^j = n_s \varepsilon_{ij} v_s^j \tag{52}$$

the force becomes

$$F_i = q_v n_s \varepsilon_{ij} (v_s^j - v_v^j) + s_v \left( E_{\omega i} - \varepsilon_{ij} v_v^j B_\omega \right). \tag{53}$$

The first term is the usual Magnus force. If $s_v \neq 0$, the term proportional to $B_\omega = \frac{1}{2} R$ acts as a curvature correction to the part of the Magnus force that depends on the vortex velocity.

Generalizing [43,44], we can compute the Magnus force from the Berry phase of a vortex describing a closed trajectory in curved space (that is asymptotically flat). The motion is along the boundary $\Gamma = \partial A$ of a neighborhood $A$ of the origin. The Berry phase is

$$\gamma_\Gamma = -\text{Im} \oint_\Gamma d\mathbf{x}_v \cdot \left\langle \Psi_v \left| \frac{\partial}{\partial \mathbf{x}_v} \Psi_v \right. \right\rangle, \tag{54}$$

with $\Psi_v$ being the many-body wavefunction for a vortex at position $\mathbf{x}_v$. The Berry phase can be written as

$$\gamma_\Gamma = \int d^2x \sqrt{g(\mathbf{x})} \oint_\Gamma dx_v^i \epsilon_{ij} \frac{\partial}{\partial x_v^j} \log|\mathbf{x} - \mathbf{x}_v| \, n_s(\mathbf{x}; \mathbf{x}_v). \tag{55}$$

Here we have generalized to curved space the expression given in [43].

Let us consider now the dual gauge field corresponding to a static density $n_s(\mathbf{x}, \mathbf{x}_v)$ with a vortex at a fixed position $\mathbf{x}_v$, the gauge potential is

$$a_i(\mathbf{x}) = -\frac{1}{2\pi} \int d^2x' \sqrt{g(\mathbf{x}')} \epsilon_{ij} \partial_j \log|\mathbf{x} - \mathbf{x}'| n_s(\mathbf{x}'; \mathbf{x}_v). \tag{56}$$

Indeed, using that

$$\partial^2 \log|\mathbf{x} - \mathbf{x}'| = 2\pi \delta^{(2)}(\mathbf{x} - \mathbf{x}'), \tag{57}$$

one gets

$$b = \frac{1}{\sqrt{g(\mathbf{x})}} \epsilon_{ij} \partial_i a_j = n_s(\mathbf{x}; \mathbf{x}_v). \tag{58}$$

Imagine now that vortices do not couple to the spin connection, i.e., $s_v = 0$. The phase shift of a vortex when it's moved around a closed path $\Gamma$ in position space is

$$\gamma'_\Gamma = -q_v \oint_\Gamma dx_v^i a_i(x_v) = \frac{q_v}{2\pi} \oint_\Gamma dx_v^i \int d^2x \sqrt{g(\mathbf{x})} \epsilon_{ij} \frac{\partial}{\partial x_v^j} \log|\mathbf{x}_v - \mathbf{x}| n_s(\mathbf{x}, \mathbf{x}_v). \tag{59}$$

Since in a bosonic superfluid the vortex charge $q_v = 2\pi$, one can see that $\gamma'_\Gamma = \gamma_\Gamma$. Therefore, the coupling to $a_\mu$ accounts for the total Berry phase and thus the coupling to the spin connection should vanish, i.e., $s_v = 0$.

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
