# Peer review of "Effective field theory of a vortex lattice in a bosonic superfluid"

_SciPost Physics, doi:SciPost Phys. 5, 039 (2018)_

## Round 2 · Referee Report · Anonymous (Referee 1) · 2018-7-16

Strengths

1) The manuscript formulates an effective field theory of a vortex lattice from the boson-vortex duality perspective. This formulation has the potential to go beyond the hydrodynamic approach.

2) The manuscript raises several very interesting questions, especially those related to the Hall viscosity, motivating the future research.

Weaknesses

1) The results on the excitation spectrum of a vortex lattice in a compressible superfluid are not new. 2) The claim of the absence of the Hall viscosity in a vortex lattice is not rigorous. However this point can be also viewed as a strength because it may motivate new research directions.

Report

Please see the attached file.

Requested changes

Please see the attached file.

Attachment

  • validity: good
  • significance: good
  • originality: good
  • clarity: good
  • formatting: good
  • grammar: good

Author:  Sergej Moroz  on 2018-09-18  [id 318]

(in reply to Report 1 on 2018-07-16)

We are grateful to the Referee for carefully reading our paper and providing valuable feedback. Below is our point-by-point replies and changes we made based on the comments of the Referee .

  1. "To make the connection between the proposed Lagrangian for a uniform system and trapped cold atomic gases in experiments more transparent, it could be useful to remind the readers that the system described by Eq.(1) can be realized when the condition $\Omega=\omega_{\perp}$ is fulfilled. Here $\omega_{\perp}$ is the transverse trapping frequency."

Authors reply: Following the Referee’s suggestion, we modified the last paragraph of the introduction section.

  1. "As far as I understood, the proposed effective field theory is valid in the vortex lattice regime (the intervortex spacing is much larger than the vortex core size) but is not applicable to the quantum Hall regime. If it is indeed the case, this should be clarified more explicitly at the very beginning of the paper."

Authors reply: We added the sentence “The effective field theory developed in this paper is not applicable in the quantum Hall regime.” in the last paragraph of the introduction section.

  1. "It could be helpful to add Anthony Zee’s book ”Quantum Field Theory in a Nutshell” as a reference about the boson-vortex duality."

Authors reply: The reference to the book of A. Zee was added, see Ref. 28.

  1. "In Eq.(1), since the current $j^{\mu}s=\varepsilon^{\mu\nu\rho} \partial\nu a_\rho$ is neutral, the source field $A_\mu$ should be a gauge field associated with an arbitrary external rotation, is it? It could be helpful if the authors can clarify this."

Authors reply: The U(1) source defined in Sec. II is associated with a deviation of the external rotation frequency from the ground state value $\Omega$. We added a short clarification in the paper after Eq. (4).

  1. "On page 6, I don’t quite understand the last sentence of the paragraph just below Eq.(5). The angular velocity Ω should change sign under P or T operation, no?"

Authors reply: The situation here is identical to the question of how an external magnetic field transforms under parity and time-reversal operations. The frequency $\Omega$ is an external parameter (i.e. it is not a field) in our effective theory and in principle remains fixed under discrete transformations. However, one can extend the time-reversal and parity transformations to external sources, in which case $\Omega\to-\Omega$ and the theory is invariant separately under the extended P and T.

  1. "In the leading order theory, Galilean symmetry is broken. What are the consequences of this? Is it a problem? Explanations are needed here."

Authors reply: In an effective field theory a generic derivative expansion just defines a kinematic regime of applicability of the theory and does not have to preserve order by order all global symmetries of a given problem. As we write in the paper, in this particular case the consequence of the breaking of Galilean symmetry in the LO approximation the vortex lattice is incompressible. On page 8 we replaced “the dual photon is instantaneous” by a more accurate statement “the dual gauge field is not dynamical”.

  1. "In the leading order theory, the Tkachenko mode that has quadratic dispersion relation is obtained. However it is also shown by the authors that to this order the vortex lattice is incompressible, namely $\partial_i u^i=0$ (above Eq.(10)). Seems to me that there is an inconsistency here. In Ref.[15], the compressibility of the vortex lattice plays a crucial role to obtain the Tkachenko mode of frequency that is quadratic in $k$. Also, intuitively, how does an incompressible vortex lattice support soft collective modes (of course no edge physics is considered here)?"

Authors reply: We believe that the compressibility of the superfluid (i.e. finite speed of sound), not compressibility of the vortex lattice fixes the Tkachenko mode to be quadratic at low momenta. An incompressible solid can support collective soft modes in the form of pure shear waves. As momentum approaches zero, the Tkachenko mode becomes strictly transverse (see Eq. 16) which is consistent with the incompressibility of the vortex lattice which we find in the LO approximation.

  1. "Eq.(12) should hold for small k. However I could not find this constraint/condition from the relevant derivations on page 8.”

Authors reply: The LO theory itself does not allow one to estimate a momentum scale where the quadratic approximation for the Tkachenko dispersion breaks down. In order to find such a scale one has to go beyond the LO theory. This is done in Sec. IIIB, where the NLO electric term is included. The Tkachenko dispersion acquires now quartic corrections, see Eq. (17). The momentum where these corrections have the same order of magnitude as the LO dispersion define the momentum scale beyond which quadratic approximation is not reliable. In the paper in order to relate better the LO and NLO calculation, we added the following sentence after Eq. (12): “In the next subsection we will find that the inclusion of the NLO electric term gives rise to quartic corrections to the Tkachenko dispersion relation."

9."Could the proposed theory cover the physics in the incompressible limit ($c_s\to\infty$), where the Tkachenko frequency should be linear in $k$?"

Authors reply: On can formally take the limit of infinite speed of sound which in our effective theory corresponds to taking the internal energy density $\varepsilon(b)$ which has an infinite curvature around the ground state value $b=n_0$. As a result, the magnetic term in the Lagrangian (8) becomes infinite leading to the constraint $\delta b=0$. Alternatively, this constraint might be implemented using a Lagrange multiplier field, but we did not pursue this direction in the present paper, where we consider only compressible superfluids.

  1. "Relevant references about the spin connection need to be added after the first sentence of section VI."

Authors reply: We added two new References 40 and 41, where the origin of the spin connection (together with its definition) in effective theories of quantum Hall fluids is discussed in detail.

  1. "A general remark: as also mentioned by the authors in the introduction, if the rotation rate $\Omega$ is very high such that the number of vortices is comparable to the number of particles, the system enters a strong correlated vortex liquid phase which can be understood as a bosonic analogy of the fractional quantum Hall sate of electrons. I would expect that this bosonic analogy exhibits a finite Hall viscosity. If the Hall viscosity is zero in a vortex lattice, as claimed by this manuscript, it could be really interesting to see how the Hall viscosity emerges as increasing the rotation rate $\Omega$ if there is any.”

Authors reply: It would be very interesting indeed to understand how the Hall viscosity changes as one melts the lattice and enters the quantum Hall regime (which exhibits a plethora of bosonic incompressible quantum Hall states). This question however goes beyond the present work which is not applicable in the quantum Hall regime.

---

## Round 2 · Referee Report · Anonymous (Referee 2) · 2018-7-17

Strengths

  1. A novel formulation of the low energy theory of the vortex lattice, which allows a systematic expansion beyond leading order.

  2. A calculation of the Hall viscosity for the vortex lattice, backed up by careful general arguments.

Weaknesses

  1. Much of the presentation involves the recovery of existing results. At times it is unclear what is new and what is known.

Report

This is an interesting paper that makes advances in the understanding of the low-energy theory of vortex lattices in compressible superfluids. While this topic is one that has a long history, the authors are clear to cite previous works which are closely related. In particular, they show that their effective theory is equivalent to that of Ref [18].

The authors emphasize that, despite this equivalence, their approach allows a systematic construction of corrections to the leading order theory. They provide descriptions of (some of) these next to leading order corrections. They also compute the response functions, and present general features expected from Galilean invariance.
Importantly, the authors calculate the Hall viscosity and argue on general grounds that it should vanish.

The paper makes useful advances in the field. In order to clarify the presentation, and the significance of their new results, I ask that the authors address the suggested changes below.

Requested changes

  1. On page 4, following Eqn (2), it would be helpful for the reader if the authors could contrast the density and current in their dual formulation with the more familiar expressions in terms of the superfluid phase (e.g. in Ref. [18]). I know that a detailed discussion of Ref. [18] is provided in the appendix. However, since page 4 is still largely introductory material, adding some targeted explanations here would be an important addition to help familiarize the reader with the meaning of the dual formulation.

  2. Please indicate more clearly which of the results are known, and to what extent the results match. For example, the text around Eqn (12) implies that the $k^2$ behavior is known; but does the prefactor not also match previous results? Do Eqns (14,16,17,18) appear in the literature? If expressions in the literature exist for these quantities, but differ in detail, please comment on this as well as the reasons for any difference.

  3. Just before Eqn (16), typo: "the the"

  4. In Section VII, the authors provide detailed arguments for the vanishing of the Hall viscosity. As they also point out in the Discussion, Sec VIII, different result applies for chiral vortex fluids. Where did the arguments of VII fail for the chiral vortex fluid? It would be helpful to insert a comment on this at the appropriate place in Sec VII.

  • validity: high
  • significance: good
  • originality: high
  • clarity: high
  • formatting: perfect
  • grammar: excellent

Author:  Sergej Moroz  on 2018-09-18  [id 317]

(in reply to Report 2 on 2018-07-17)

We are grateful to the Referee for carefully reading our paper and providing valuable feedback. Below is our point-by-point replies and changes we made based on the comments of the Referee.

  1. The structure of our paper makes it difficult to discuss the relation between the Watanabe&Murayama and dual formulations already after Eq. (2). To make our presentation more coherent we added on page 5 the following sentence: “Moreover, the dual electric and magnetic fields are related to the regular part of the superfluid phase and displacement vectors via Eqs. (2) and (A3).”

  2. The prefactor in Eq. (12) matches previous results and to emphasize it we reformulated the sentence after Eq. (12). To the best of our knowledge Eq. (14) is new. As for the NLO results (16,17), we did not attempt to make a detailed comparison with the existing literature because our NLO theory might be not the most general one. This is emphasized in the paper in the last paragraph of Sec III.

  3. The typo is corrected.

  4. A chiral vortex fluid that arises in an incompressible fluid studied in Refs. [46,47] is a different phase of matter from the vortex lattice in compressible superfluid investigated in the present paper and thus a priori the Hall viscosities do not have to match. Note, however, that since our NLO theory is not complete, we prefer not to make strong claims about the value of the Hall viscosity (see the last sentence of the abstract and the last paragraph of Sec. VI) and relation to the findings of Refs [46, 47].

---

## Round 3 · Referee Report · Anonymous · 2018-9-24

Report
The authors have responded to all of the points raised in my first report. In view of the strengths listed there, and the subsequent revisions, I am happy to recommend the paper for publication in its current form.

---

## Round 3 · Referee Report · Anonymous · 2018-9-25

Report
The authors have addressed carefully all the points raised in
my previous report and modified the manuscript accordingly. I support the publication in the current form.

---

## Round 3 · Author Response

Thank you for considering our paper for publication in SciPost.
We are grateful to both Referees for carefully reading our paper and providing valuable feedback. In two replies we provided point-by-point answers to the comments of the Referees and the list of changes implemented in the present resubmission.
with best regards,
Sergej Moroz, Carlos Hoyos, Claudio Benzoni and Dam Thanh Son

You are currently on this page

---

## Editorial Decision

published